# The DNA co-vaccination using Sm antigen and IL-10 as prophylactic experimental therapy ameliorates nephritis in a model of lupus induced by pristane

Beatriz Teresita Martín-Márquez[1,2], Minoru Satoh[3], Rogelio Hernández-Pando[4], Erika Aurora Martínez-García[1,2,5], Marcelo Heron Petri[1,6], Flavio Sandoval-García[1,7,8], Oscar Pizano-Martinez[1,2,9], Trinidad García-Iglesias[10], Fernanda Isadora Corona-Meraz[1,8,11], Monica Vázquez-Del Mercado[1,2,12]*

1 Departamento de Biología Molecular, Instituto de Investigación en Reumatología y del Sistema Músculo Esquelético (IIRSME), Centro Universitario de Ciencias de la Salud, Universidad de Guadalajara, Guadalajara, Jalisco, México, 2 Departamento de Biología Molecular, UDG-CA-703, "Inmunología y Reumatología", Guadalajara, Mexico, 3 Department of Clinical Nursing, School of Health Sciences, University of Occupational and Environmental Health, Kitakyushu, Fukuoka, Japan, 4 Departamento de Patología, Sección de Patología Experimental, Instituto Nacional de Ciencias Médicas y Nutrición Salvador Zubirán, Ciudad de México, México, 5 Departamento de Fisiología, Centro Universitario de Ciencias de la Salud, Universidad de Guadalajara, Guadalajara, Jalisco, México, 6 Department of Cardiothoracic and Vascular Surgery, Örebro University Hospital, Örebro, Sweden, 7 Departamento de Neurociencias, Centro Universitario de Ciencias de la Salud, Universidad de Guadalajara, Guadalajara, Jalisco, México, 8 Departamento de Biología Molecular, UDG-CA-701, "Envejecimiento, Inmunometabolismo y estrés oxidativo", Ciudad de La Habana, Cuba, 9 Departamento de Morfología, Centro Universitario de Ciencias de la Salud, Universidad de Guadalajara, Guadalajara, Jalisco, México, 10 Departamento de Fisiología, Laboratorio de Inmunología, Centro Universitario de Ciencias de la Salud, Universidad de Guadalajara, Guadalajara, Jalisco, México, 11 División de Ciencias de la Salud, Departamento de Ciencias Biomédicas, Centro Universitario de Tonalá, Universidad de Guadalajara, Tonalá, Jalisco, México, 12 División de Medicina Interna, Hospital Civil "Dr. Juan I. Menchaca", Servicio de Reumatología PNPC 004086 CONACyT, Guadalajara, Jalisco, México

* dravme@hotmail.com

## Abstract

### Introduction

Systemic lupus erythematosus (SLE) is an autoimmune disease characterized by the production of autoantibodies such as anti-Sm. Studies in patients with SLE and murine models of lupus reveal that the most critical anti-Sm autoantibodies are predominantly direct against $D1_{(83-119)}$, D2, and B´/B epitopes.

### Objectives

The present study aimed to analyze the induction of antigen-specific tolerance after prophylactic immunization with a DNA vaccine encoding the epitopes: $D1_{83-119}$, D2, B´/B, and B´/$B_{COOH}$ in co-vaccination with IFN-γ or IL-10 in a murine model of lupus induced by pristane.

### Material and methods

To obtain endotoxin-free DNA vaccines, direct cloning techniques using pcDNA were performed: $D1_{83-119}$, D2, B´/B, B´/$B_{COOH}$, IFN-γ, or IL-10. Lupus was induced by 0.5 mL of

**Data Availability Statement:** All relevant data are within the manuscript and its Supporting Information files.

**Funding:** MVM Grant 51353 supported by SEP-CONACyT 51353 grant from Consejo Nacional de Ciencia y Tecnología (CONACyT). URL:http://www.conacyt.gob.mx The funders had no role in study design, data collection and analysis, decision to publish, or preparation of the manuscript.

**Competing interests:** The authors have declared that no competing interests exist.

pristane via intraperitoneal in *BALB/c* female mice. Immunoprecipitation with K562 cells was metabolically labeled with $^{35}S$ and ELISA to detect serum antibodies or mice IgG1, IgG2a isotypes. ELISA determined IL-10 and IFN-γ from splenocytes supernatants. Proteinuria was assessed monthly, and lupus nephritis was evaluated by immunofluorescence, and electron microscopy.

## Results

The prophylactic co-vaccination with D2/IL-10 reduced the expression of kidney damage observed by electron microscopy, direct immunofluorescence, and H & E, along with reduced level of anti-nRNP/Sm antibodies (*P* = 0.048).

## Conclusion

The prophylactic co-vaccination of IL-10 with D2 in pristane-induced lupus ameliorates the renal damage maybe by acting as prophylactic DNA tolerizing therapy.

## Introduction

Systemic lupus erythematosus (SLE) is an autoimmune disease characterized by the spontaneous production of autoantibodies against self-antigens such as double-stranded DNA (dsDNA), histones, and small nuclear ribonucleoproteins (snRNP), including Smith (Sm) antigen. The anti-Sm antibodies are considered for the European League Against Rheumatism (EULAR/American College of Rheumatology (ACR) as a classification criterion for SLE diagnosis [1–3]. SLE patients present various clinical manifestations resulting from multiple-organ damage, especially the skin, kidney, heart, and central and peripheral nervous system, among others [4]. Anti-Sm antibody was first detected in 1966 and is present in approximately a third of SLE patients associated with nephritis [5–7]. The Sm antigen is part of the spliceosomal complex and is composed of at least nine different polypeptides with molecular weights ranging from 9–29.5 kDa, including B (B1, 28), B´ (B2, 29), N (B3, 29.5), D1 (16), D2 (16.5), D3 (18), E (12), F (11) and G (9). The Sm core complex is associated with U1, U2, U4, U5 RNA and participates in pre-mRNA splicing [6, 8]. Autoantigenicity against Sm polypeptides has been detected for most of them; nevertheless, the primary targets are B and D1 polypeptides [5, 8, 9]. Regarding D1 polypeptide, studies revealed that the C-terminal peptide of D1 is a glycine-arginine (GR)-rich epitope (residues 83–119) that is recognized as a significant target and is associated with human and murine lupus with 70% of sensitivity and 94% of specificity [8, 10–12]. Meanwhile, about B´/B polypeptide, an octapeptide PPPGMRPP repeated three times at the carboxy terminus (B´/B$_{COOH}$), is a target of the early autoimmune response in some SLE patients [13]. Experimental studies demonstrate that immunization with PPGMRPP induces lupus, and it has been hypothesized that this octapeptide has a structural similarity with the Epstein-Barr virus (EBV) nuclear antigen (EBNA-1) [14]. On the other hand, it has been identified that the D2 polypeptide could play a role in SLE because specific D2 epitopes are involved in partly binding D1 90–102 amino acids, being one of the highest areas of reactivity in D2-positive SLE sera [12]. However, there is no information about a specific antigen induction or protective role related to the D2 epitope in murine models of lupus.

During the development of nacked DNA vaccine therapy in experimental lupus, one approach could be using of a tailoring therapy using co-vaccination: autoantigen epitopes and cytokines could be a promising clinical approach that, in the best scenario, would be able to

translate it from bench to bedside. A prototype cytokine that meets all the experimental requirements to be a key molecule in triggering kidney damage in mice lupus is the IFN-γ, a $T_H1$ cytokine [15–17]. IFN-γ is considered crucial in the pathogenesis of lupus nephritis based on genetically susceptible strains like NZB x W (B x W) F1 [18]. Notwithstanding, it has been observed that continuous administration of anti-IL-10 ($T_H2$) delays the onset of autoimmunity in B x W mice [19] with a suppressive effect of IL-10 in genetically deficient mice of IL-10 MRL-Fas$^{lpr}$. These findings suggest that IL-10 can regulate murine lupus, particularly in early stages, and can act as an enhancer or inhibitor of autoimmunity, altering the penetrance of autoantibodies and the production of IgG2a [20].

It has been proven that DNA co-vaccination along with some autoantigens $T_H1$ or $T_H2$ cytokines might be a potential tool used for experimental treatment in autoimmune diseases such as rheumatoid arthritis (RA), experimental allergic encephalomyelitis (EAE) and, autoimmune diabetes [21, 22]. In this study, we aimed to evaluate the induction of antigen-specific tolerance by a prophylactic co-vaccination with DNA encoding Sm antigens along with IL-10 or IFN-γ in a pristane-induced murine model of lupus.

## Materials and methods

### Animals

A total of 130 female *BALB/c* mice from 6–8 weeks old were obtained from Universidad Nacional Autónoma de México-Envigo RMS Laboratory in México City, and housed in the animal facility of Instituto de Investigación en Reumatología y del Sistema Músculo Esquelético, Centro Universitario de Ciencias de la Salud, Universidad de Guadalajara, under the following conditions: 2–4 animals in clear cages (7.6x11.6x4.8 inches) classified by a numerical and color code per treatment, controlled temperature room at 22±1˚C, positive laminar flow, 40–70% relative humidity, 12 hours of light/dark cycles, pine shavings bedding and fed *ad libitum* with purified water and with normo-caloric diet (Harlan Tekland™ Global 2019). The prophylactic and control groups were randomly assigned. In efforts to ensure animal welfare and alleviate the suffering, we established the following procedures: single trained operator for animal allocation and handling during the stages of the experiment, sufficient bedding material (1 cm depth), assurance sleep/wake cycles, light levels not greater than 300 lumens and noise no more than 85dB [23]. Since this was a prophylactic approach, we do not include inclusion criteria beyond the strain, gender weeks of age, or exclusions for experiments.

Mice were monitored weekly, and we established an ethical endpoint (proteinuria plus deteriorating general health condition, weight loss > 20%, immobility greater than 12 hr., epilepsy, coma, paralysis, and extreme pain). The measure of spontaneous emitted pain was assessed with the mouse grimace scale [24]. Mice were euthanized by $CO_2$ inhalation in a euthanasia chamber at a rate of 20–30% $CO_2$ chamber volume per minute. Blood, spleen, and kidneys were collected for further analysis. The Institutional Review Board approved the protocol (IRB) of the Centro Universitario de Ciencias de la Salud, Universidad de Guadalajara (MX/a/2016/010823) and, all experimental procedures were carried out in compliance with the guidelines for animal research (NOM 0062-ZOO-1999 and NOM-033-ZOO-1995) and were performed to analyze the induction of antigen-specific tolerance after prophylactic immunization with a DNA vaccine encoding the Sm epitopes in co-vaccination with IFN-γ or IL-10 in a murine model of lupus induced by pristane.

### Tolerizing DNA vaccines

DNA constructs encoding murine Sm antigens as D1$_{83-119}$; D2; B′/B; B′/B$_{COOH}$ and the cytokines IFN-γ and IL-10 were cloned into pcDNA™ 3.1 D/V5-His-TOPO® mammalian

expression vector (Invitrogen™, Carlsbad, CA). cDNA encoding for these molecules was amplified from mouse splenocytes using RT-PCR and the following oligonucleotides primers: $D1_{83\text{-}119}$ forward 5′ `CACCGTTGAACCTAAGGTGAAGTCTAAGAAA` 3′ and reverse 5′ `TCGCCTAGG` `ACCCCCTCTTCCTCTGCC` 3′; $D2$ forward 5′ `CACCATGAGTCTCCTCAATAAACCC` 3′ and reverse 5′ `CTTGCCAGCGATGAGTGGGTTCCG` 3′, full-length $B′/B$ forward 5′`CACCA` `TGACGGTGGGCAAGAGCAGCAAGATGCTG` 3′ and reverse 5′ `AGGCAGACCTCGCATGCCT` `GGAGGAGG` 3′; $B′/B_{COOH}$ forward 5′ `CACCCCCCCTCCTGGCATGCGGCCTCCT` 3′ and reverse 5′ `AGGAGGCCGCATGCCAGGAGGGGG` 3′; $IFN\text{-}\gamma$ forward 5′ `CACCATGAACGCTA` `CACACTGC` 3′ and reverse 5′ `GCAGCGACTCCTTTTCCGCTTCCT` 3′; $IL\text{-}10$ forward 5′ `CACCTAGAGACTTGCTCTTGCACTACCAA` 3′, and reverse 5′ `ATCCCTGGATCAGATTTAG` `AGAGCT` 3′. The RT-PCR was performed in a final reaction volume of 50μL (30μM forward and reverse primer, 10X PCR Buffer, $Taq$ DNA polymerase 5U/μL and, 2μL cDNA). The reaction conditions were: holding at 95°C/4 min, cycling at 30 cycles of 95°C/45s, 60°C/90s and, 72°C/90s. DNA constructs were amplified in $Escherichia\ coli$ One Shot® Top10 (Invitrogen™, Carlsbad CA), then we obtained the DNA of interest using Qiagen Endo-Free® Giga Prep kits (Qiagen™ GmbH D-40724 Hilden).

## Prophylactic treatment

The experimental animals were divided into eight groups of 13 female $BALB/c$ mice per treatment and received an injection of 0.1 mL of 0.25% bupivacaine-HCl (Sigma-Chemical Co., St Louis, MO, USA) in DPBS (Gibco® Invitrogen™) in the left quadriceps at day 0. On days 2 and 10, the mice received intramuscular injections of 100 μg of a cocktail mixture of pcDNA™3.1 D/V5-His-TOPO® ™ (Invitrogen™ Carlsbad, CA) encoding either: $D1_{83\text{-}119}$, D2, B′/B or B′/B$_{COOH}$ co-vaccinated IFN-γ or IL-10 encoded into pcDNA™3.1D/V5-His-TOPO® (Fig 1). The sample size was calculated based on tolerizing DNA vaccines designed experiments [21, 25], and the exact value of $n$ in each experiment was described in Fig 1.

## Control group

Thirteen female $BALB/c$ mice conformed to the mock treatment. The animals were injected with 0.1 mL of 0.25% bupivacaine-HCl (Sigma-Chemical Co., St Louis, MO, USA) in the left quadriceps at day 0 and on day 2 and 10 received an intramuscular injection of 100μg of empty vector pcDNA™3.1 D/V5-His-TOPO®. 13 female $BALB/c$ mice received 0.1 mL of Phosphate Buffer Solution (PBS; Gibco® Invitrogen™) for the control group (Fig 1).

## Lupus induced by pristane

Experimental animals received a single intraperitoneal injection of 0.5 mL of pristane (2,6,10,14-tetramethylpentadecane; Sigma-Chemical Co., St. Louis, MO, USA) on day 16 after DNA immunization (Fig 1) [26].

## Immunoprecipitation

The presence of autoantibodies was tested by immunoprecipitation of $^{35}$S-methionine labeled K562 (human erythroleukemia) cell extract and SDS-PAGE as described. Briefly, the cells were labeled with [$^{35}$S] methionine (DuPont-New England Nuclear, Boston, MA), lysed in NET/NP-40 buffer (150 mM NaCl, 2mM EDTA, 50 mM Tris-HCl pH 7.5, 0.3% NP-40), containing 0.5 mM PMSF, 0.3 TIU/mL aprotinin, and immunoprecipitated using protein A-Sepharose beads (Pharmacia LKB Biotechnology, Inc, Piscataway, NJ) coated with 5 μL of mouse sera plus 12 μL of rabbit anti-mouse IgG1 (1 mg/mL). After, several washes were done using 0.5 M

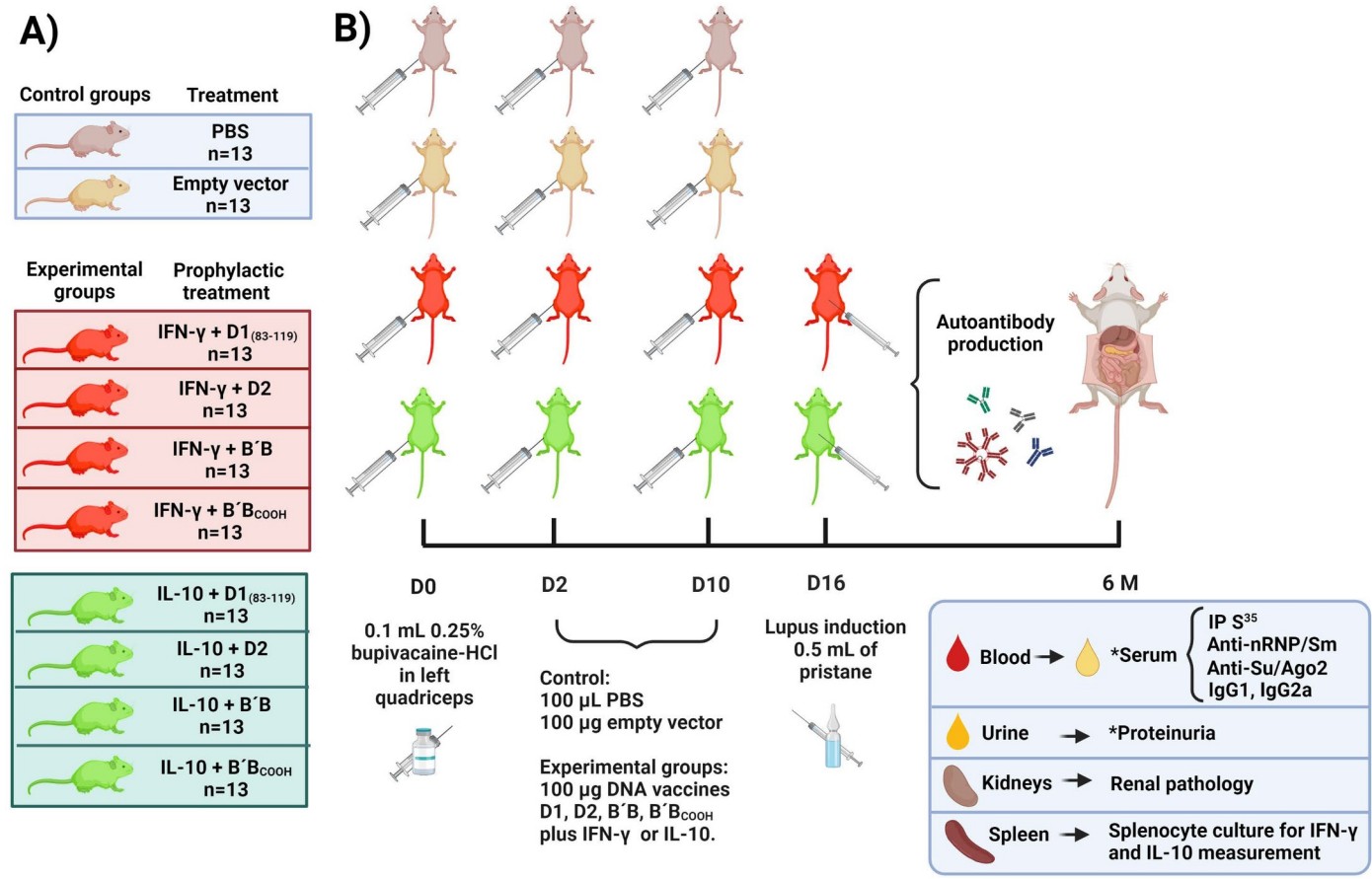

**Fig 1. Schematic representation of the study design of prophylactic DNA vaccination in a murine lupus model.** Panel A) Female *BALB/c* mice were included as control group and experimental group. Panel B) Experimental procedures timepoint for control group and prophylactic DNA vaccination group. Abbreviatures: D = day, M = month, IP = immunoprecipitation, HCl = hydrochloric acid, S = sulfur, PBS = Phosphate Buffer Solution. *In these determinations, some samples run out. Created with BioRender.com.

NaCl NET/NP40 (0.5 M NaCl, 2 mM EDTA, 50 mM Tris-HCl, pH 7.5, 0.3% NP-40). American Type Culture Collection, Rockville, MD). Immunoprecipitated proteins were analyzed by autoradiography [27].

## Anti-nRNP/Sm ELISA

Anti-nRNP/Sm antigen-capture ELISA was performed as described. Briefly, microtiter plates (Nunc, Immobilizer Amino™) were coated with 3 µg/mL mouse monoclonal antibodies (mAb) 2.73 (IgG2a, antiU1-70k). The left half of the plate was incubated with K562 cell lysate (50 µL/well, 4 X $10^7$/mL), and the right half was incubated with the blocking buffer as control. After washing the plate, the same set of samples and serially diluted standard serum (1:500 to serial 1:5 dilutions) were added to the plate´s left and right half (control for reactivity against mouse IgG).

Serum samples were tested at 1:500, and 1:2,500 dilution and data from the latter was used for the analysis. Plates were washed with TBS/Tween20, incubated with alkaline phosphatase (ALP)-conjugated mouse mAbs to human IgG (Sigma Aldrich™, 1:1,000 dilution), and developed. The optical density (O.D.) 405nm of wells were converted into units based on the standard curve, and the units of the corresponding right half (without nRNP/Sm antigens) were

subtracted from the left half (with antigens) using SoftMax Pro 4.3 software (Molecular Devices, Sunnyvale, CA, USA).

### Anti-Su/Ago2 ELISA

The determination of anti-Su/Ago2 by ELISA was as described. Wells of microtiter plates (Nunc, Immobilizer Amino™) were coated with 50 μL of human anti-Su IgG at 20 μg / mL in 20 mM Tris-HCl pH 8.0. Half of the wells were incubated for 1 h at 22˚C with 50 μL of K562 cell lysate and the other half with NET/NP-40 alone, 100 μL of diluted mouse serum (1/250 in blocking solution) was added to wells coated with either K562 cell lysate of buffer alone, and incubated for 1 h at room temperature. The wells were then washed using with NET/NP-40, and incubated with alkaline phosphatase-conjugated goat anti-mouse IgG antibodies (Southern Biotechnology, Birmingham, AL), later developed with 1 mg / 1mL $p$-nitrophenyl phosphate substrate (Sigma) in diethanolamine buffer (1 M diethanolamine-HCl pH 9.6, 0.5 nM $MgCl_2$, 0.02% $NaN_3$) and OD was read at 405 nm in an automated ELISA reader [28].

### Measurement of immunoglobulin (IgG) levels

Total levels of each Ig isotype were determined by ELISA coated with 50 μL/well of 3 μg/mL of goat anti-mouse κ/λ light chain antibodies (9:1 ratio, Southern Biotechnology, Birmingham, AL). Wells were washed with 0.15 M NaCl, 2 mM EDTA, 50 mM Tris containing 0.3% (NET/NP-40) and then blocked with 0.5% BSA in NET/NP-40. Murine sera were diluted 1:200,000 with NET/NP-40 containing 0.5% BSA. The plate was washed and incubated with 1:1000 dilution of alkaline phosphatase-labeled goat anti-mouse antibodies specific for IgG1 and IgG2a (Southern Biotechnology). Read at 405 nm using an ELISA plate reader [29].

### Splenocytes culture and cytokine measurement

In a sterile Petri dish, a pool of 4 spleens of control or experimental mice were extracted and placed in 5 mL of Hank´s balanced salt solution (Sigma-Aldrich™). The splenocytes were separated with Hystopaque™ (Sigma-Aldrich™) in a 2:1 ratio. The button was resuspended in 1 mL of RPMI 1640 non-supplemented medium (Gibco® Invitrogen™), and the cell count was performed in a Neubauer chamber with 10μL of trypan blue and 10μL of cell suspension. Briefly, the cell suspension was placed in 5 mL of RPMI 1640 (Gibco® Invitrogen™) supplemented with penicillin/streptomycin at 1% (Gibco® Invitrogen™) in a culture flask and was incubated 24 hours at 37˚C. Culture supernatants were harvested and stored at -20˚C. Cytokine levels of IFN-γ and IL-10 in culture supernatants were measured by ELISA (Mouse Biotrak ELISA System GE Healthcare™).

### Proteinuria

Urine samples were collected by spontaneous urination at six months after lupus was induced by pristane. For semi-quantitative measurement of proteinuria, Multistix® urinalysis strips (Bayer®) were employed. According to the color scale provided by the manufacturer, albuminuria was categorized as follows: 0–1 = trace, 1 = 30, 2 = 100, 3 = 300 and 4 > 2000 mg/dL.

### Renal pathology

Immediately after euthanized the mice, both kidneys from three animals per experimental group were removed and divided into two halves. For immunofluorescence study, one-half was frozen by immersion in liquid nitrogen and sectioned at 5 microns thickness, incubated with FITC-conjugated rabbit anti-mouse IgG antibodies, and analyzed under an

epifluorescence microscope. From the other kidney half, we obtained a thin layer of 1mm from the cortex, fragmented into small tissue pieces immediately fixed by immersion in 4% glutaraldehyde dissolved in cacodylate buffer (Sigma Aldrich™) for 2hr at 4°C. After washing, kidney tissue fragments were dehydrated in graded ethyl alcohols and embedded in Epon resin. Sections of 1-micron thickness were stained with toluidine blue and used to select representative tissue areas from which thin sections were obtained. Thin tissue sections were mounted in copper grids, stained with citrate lead and uranium salts and examined in a Zeiss EM 10 electron microscopy. The rest of the kidney tissue was fixed by immersion in 10% formaldehyde dissolved in PBS, dehydrated, and embedded in paraffin for conventional histology.

## Statistical analysis

We used the Kolmogorov-Smirnov test to determine the distribution of the data. After considering that only a non-parametric test can be used, comparisons were made using Kruskal-Wallis, Wilcoxon rank-sum test, Mann-Whitney U as applicable. All data were analyzed using statistical software packages IBM SPSS Statistics v24 (IBM Inc., Chicago, IL, USA) and Graph-Pad Prism v6.01 (2014 Inc. 2236 Beach Avenue Jolla, CA 92037). Statistical significance was obtained when $P \leq 0.05$.

## Results

### IFN-γ dependence of anti-RNP/Sm antibodies

Autoantibodies in sera from experimental prophylactic co-vaccinated groups were tested by immunoprecipitation 6 months after pristane treatment and we observed the following prevalence for anti-nRNP/Sm autoantibodies: IFN-γ/D1$_{83-119}$ (27%) *vs.* IL-10/D1$_{83-119}$ (25%); IFN-γ/D2 (54%) *vs.* IL-10/D2 (10%); IFN-γ/B´B (30%) *vs.* IL-10/B´B (0%) and IFN-γ/B´/B$_{COOH}$ (18%) *vs.* IL-10/B´/B$_{COOH}$ (36%). Although we did not obtain significant differences, these data suggest that the prophylactic co-vaccination with IFN-γ and pristane treatment enhance the autoantibody production in an IFN-γ dependence manner (Fig 2). All SDS PAGE images are available in S1 File.

The prevalence for anti-Su/Ago2 autoantibodies were alike: IFN-γ/D1$_{83-119}$/(27%) *vs.* IL-10/D1$_{83-119}$ (16%); IFN-γ/D2 (18%) *vs.* IL-10/D2 (30%); IFN-γ/B´/B (20%) *vs.* IL-10/B´B (23%) and IFN-γ/ B´/B$_{COOH}$ (9%) *vs.* IL-10/B´/B$_{COOH}$ (9%).

Next, we also assessed ELISA anti-nRNP/Sm and anti-Ago2/Su antibodies and observed differences in anti-nRNP/Sm titers between the co-vaccinated groups with D2/IFN-γ *vs.* D2/IL-10 ($P = 0.048$) (Fig 3). In other words, the IFN-γ dependence was reliable when we tested by ELISA in the group of D2 co-vaccinated with IFN-γ.

### IgG1, IgG2a subclass concentration, and IgG2a/IgG1 ratio

We did not obtain statistically significant differences in the levels of IgG1, IgG2a, and IgG2a/IgG1 ratio between the groups vaccinated with IL-10 and IFN-γ (Fig 4).

### Proteinuria

We analyzed the proteinuria levels in the co-vaccinated groups six months after pristane treatment, and we observed that significant proteinuria was less common in IL-10 *vs.* IFN-γ (Fig 5).

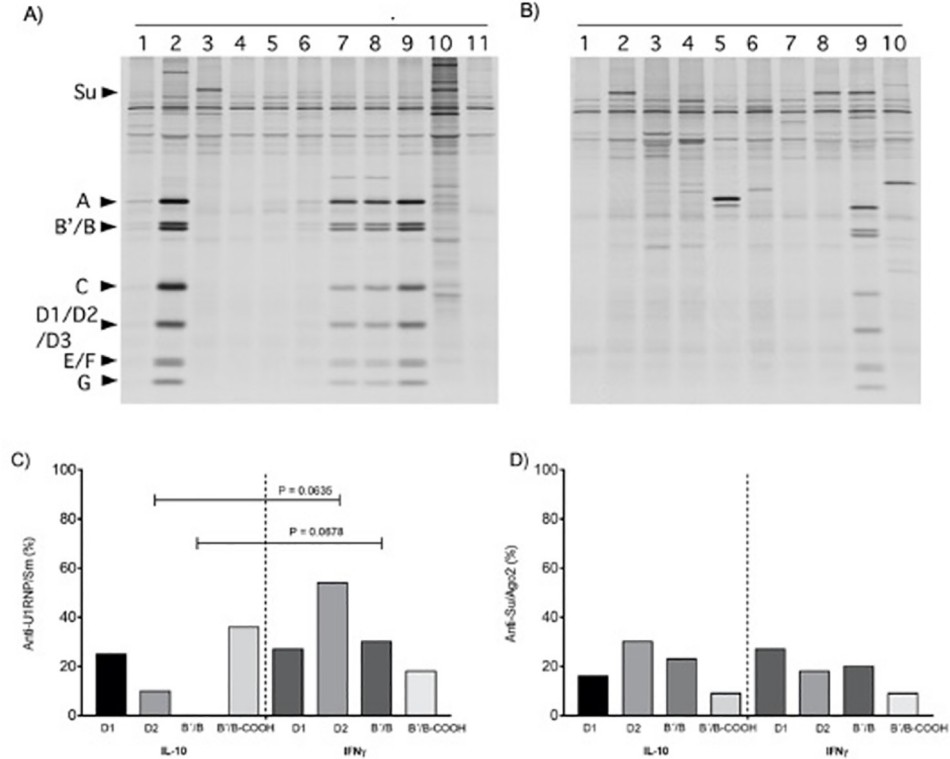

**Fig 2. Immunoprecipitation using sera from pristane-treated mice co-vaccinated with D2 plus IFN-γ vs. IL-10.**
$^{35}$S-methionine-labeled K562 cell extract was immunoprecipitated using sera from pristane-treated mice. Panel A)
IFN-γ/D2 co-vaccinated group: anti-nRNP/Sm (A, B′/B, C, D1/D2/D3, E/F, and G) antibodies are observed in lanes
2,7,8 and 9, and anti-Su (100kDa) in lane 3 and 10. Panel B) IL-10/D2 co-vaccinated group: anti-nRNP/Sm (A, B′/B,
C, D1/D2/D3, E/F, and G) is observed in lane 9 and anti-Su antibodies in lanes 2,8, and 9. Panel C) Prevalence of anti-
nRNP/Sm antibodies. Panel D) Prevalence of anti-Su/Ago2 antibodies. The results are shown in percentage (%,
Wilcoxon rank-sum test).

## Cytokine concentration in splenocyte supernatants

We analyzed the mouse IFN-γ and IL-10 splenocyte supernatant levels in control mice con-
formed by PBS and mock treatment and in prophylactic co-vaccinated treated groups D1$_{83-}$
$_{119}$, D2, B′/B and B′/B$_{COOH}$ along with IFN-γ and IL-10. In PBS and mock-treated control
mice, we observed that the IL-10 or IFN-γ levels were non-significant (Fig 6A). Nevertheless,
we found a substantial increase in IFN-γ concentrations from splenocytes supernatants of pro-
phylactic co-vaccinated groups treated with IFN-γ, obtaining significant differences between
D1$_{83-119}$ vs. D2 ($P = 0.001$), D1$_{83-119}$ vs. B′/B ($P = 0.045$) and D1 vs. B′/B$_{COOH}$ ($P = 0.001$)
(Fig 6A). In Fig 6B the levels of splenocytes supernatant of IL-10 were not different between
the experimental groups.

## Renal pathology findings

The light microscopy study of the kidneys from mice treated with D2/IFN-γ showed more
than 90% of glomeruli, accentuated/diffuse increase of mesangial matrix, cellularity with capil-
lary thickening and splitting as the accentuation of lobular architecture (Fig 7A). The immu-
nofluorescence study showed numerous IgG immune complexes deposited in the mesangial
and capillary walls with intramembranous, subendothelial/subepithelial locations (Fig 7B).
The electron microscopy study confirmed these features; moreover, it allowed us to enlarge

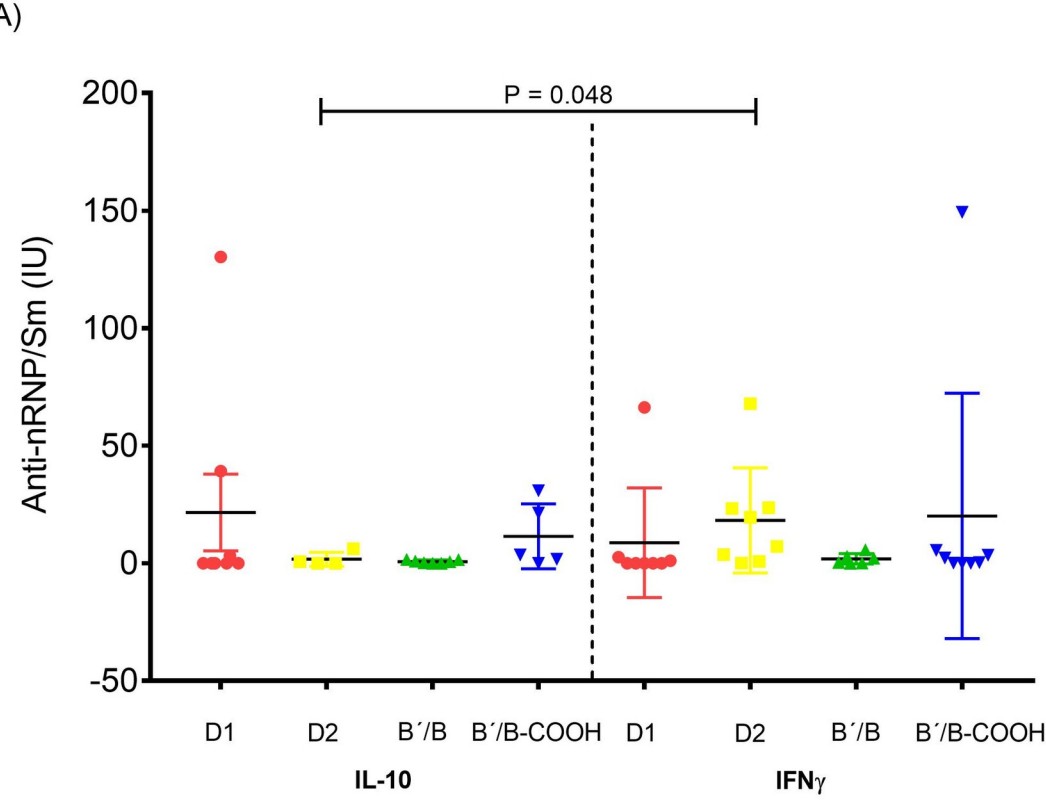

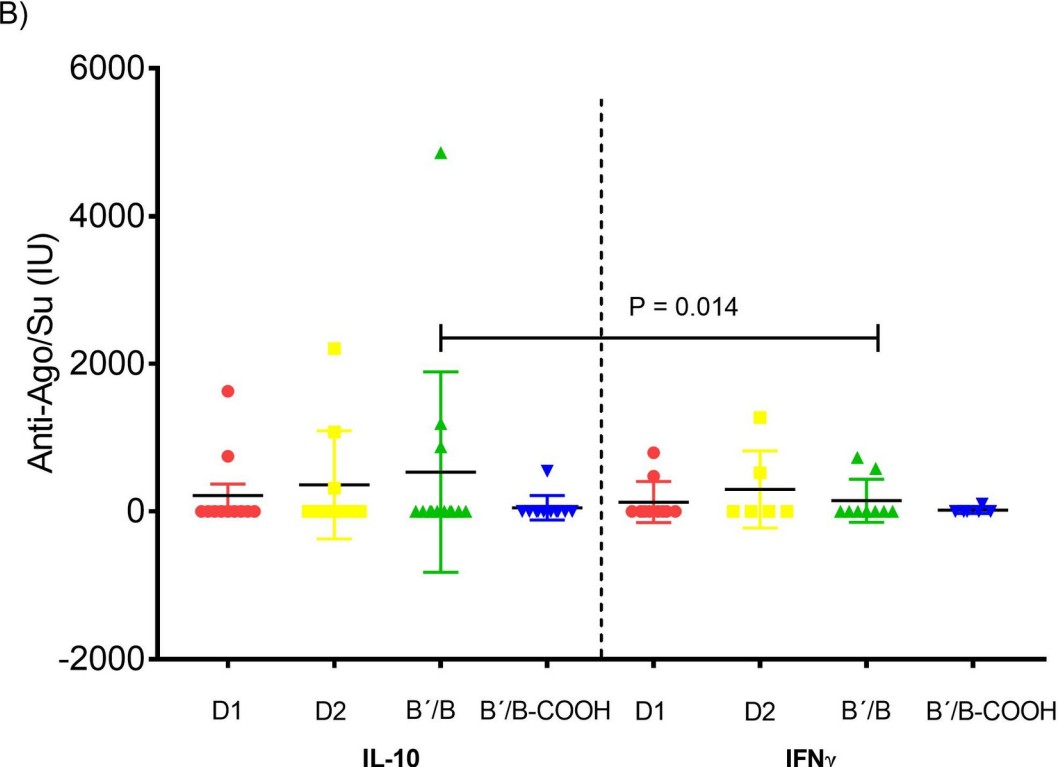

**Fig 3. Effects of prophylactic co-vaccination of Sm antigens with IL-10 or IFN-γ on anti-nRNP/Sm and anti-Su/Ago2 antibodies level.** Effects of prophylactic co-vaccination of Sm antigens with IL-10 or IFN-γ on anti-nRNP/Sm and anti-Su/

Ago2 antibody production. Sera were tested for Panel A) anti-nRNP/Sm and Panel B) anti-Su/Ago2 antibodies by ELISA at six months after prophylactic co-vaccination and induction of lupus by pristane. Each symbol represents one mouse. Mann-Whitney test was used.

our findings showing edema of podocytes with partial fusion of pedicels and numerous electron-dense deposits essentially intramembranous and in lower amounts subendothelial and subepithelial (Fig 7C). These changes were strikingly less apparent in the kidneys from IL-10/D2 vaccinated animals (Fig 7D–7F). The images of the histology of kidneys from control and prophylactic DNA vaccination in a murine lupus model are available in S1 Fig.

## Discussion

SLE is an autoimmune disease characterized by a chronic immune response against cells, tissues, and organs related to loss of self-tolerance to self-antigens that might be triggered in a susceptible individual by environmental, genetic, or other factors [30, 31]. In other rheumatic diseases such as RA, there are plenty of targets sensitive for biological treatment that has proved effective to achieving a rapid and sustained clinical remission, as stated in the Treat to Target strategy (T2T) [32].

It is desirable in SLE to have multiple targets for a successful treatment, especially when a major organ is involved, such as renal damage (lupus nephritis). However, only belimumab has been approved by the Food and Drug Administration (FDA), and according to the BLISS (Belimumab in Subjects with Systemic Lupus Erythematosus) trial is not effective in lupus nephritis [33]. This study is promising as a prophylactic tailored experimental lupus nephritis therapy, using the approach of self-epitopes derived from Sm antigen, co-vaccinated using opposite cytokines: the critical molecule in lupus nephritis IFN-γ vs. IL-10 [34]. The murine model of lupus induced by pristane, is an acknowledged model with the importance of environmental factors that might predispose to SLE development [31]. The single intraperitoneal injection of pristane can induce in female *BALB/c* mice a wide range of autoantibodies against the RNA component of U1 snRNP through a T-cell dependent immune response together with mild glomerulonephritis and arthritis [31]. This model could be suitable analyzing of promising therapies based on the development of tolerance against Sm peptides.

In the present study, we conducted a prophylactic co-vaccination with DNA plasmids encoding $D1_{83-119}$, D2, B´/B, and B´/$B_{COOH}$ epitopes and IL-10 or IFN-γ with a follow-up period of 6 months after lupus induced by pristane. We designed DNA vaccines based on the antigenicity of $D1_{83-119}$ and, B´/B polypeptides demonstrated to induce autoimmunity in experimental models. In addition, we included the D2 polypeptide to describe its antigenicity. The polypeptide $D1_{83-119}$ is a vital T cell epitope in SLE patients [11] with high sensitivity and disease specificity [8]. The B´/B peptide, it has been observed that the three repeats of a proline-rich peptide sequence (PPPGMRPP) in the C-terminus could trigger an autoimmune response in immunizing rabbits [35]. These rabbits developed SLE symptoms, clinical features of lupus, and B cell spreading of the autoimmune reaction to B´/B, and D1, U1-70K, U-A, and U1-C [13, 35]. The D2 epitope has been and has an isoelectric point significantly higher (basic), especially in the residues 83–105, which could be antigenic [12].

To analyze the prophylactic effects of co-vaccination of $D1_{83-119}$, D2, B´/B, and B´/$B_{COOH}$ along IFN-γ and IL-10, we tested the sera to determine the prevalence of anti-U1RNP/Sm antibodies by immunoprecipitation. We noted that, although we did not obtain differences in the D2/IL-10 co-vaccinated group, we observed differences in the frequency of anti-U1RNP/Sm vs. D2/IFN-γ co-vaccinated group (10% vs. 54%, respectively) (Fig 1B). In addition, the anti-Su/Ago2 antibodies were higher in the group co-vaccinated with D2-IL-10. Original studies

A)

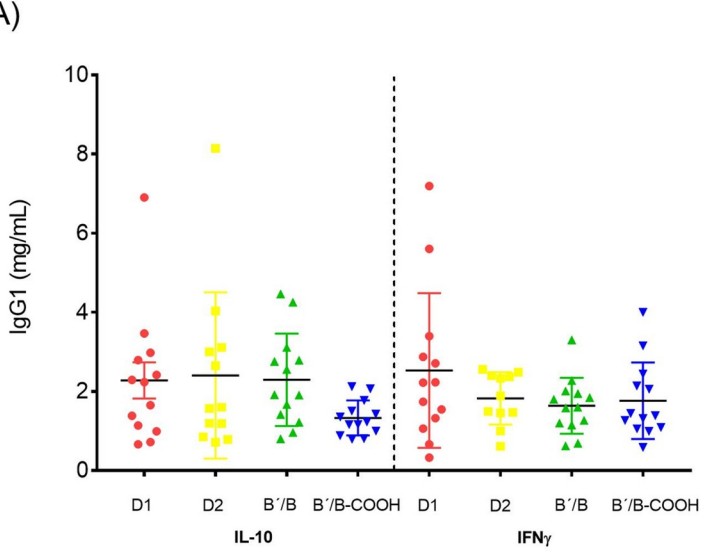

B)

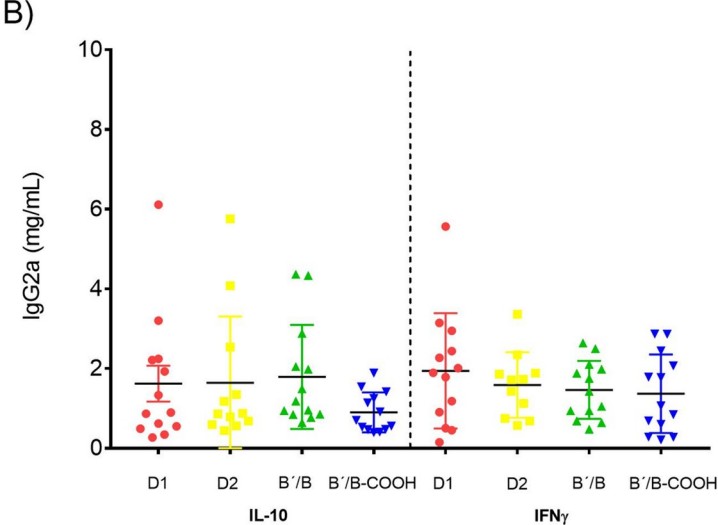

C)

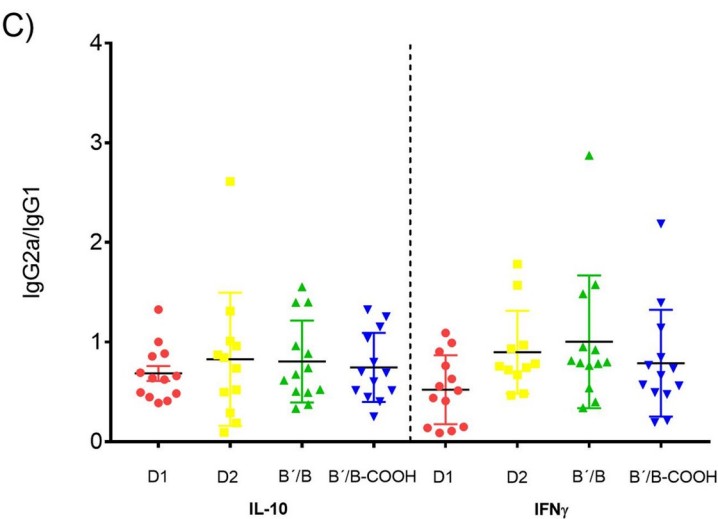

**Fig 4. Levels of serum IgG1, IgG2a and IgG2a/IgG1 ratio subclass by ELISA.** Each symbol represents one mouse. Mann-Whitney test was used.

describing lupus induced by pristane in *BALB/c* female mice showed prevalence for anti-nRNP/Sm by about 53 to 78% [15, 29, 36] and for anti-Su/Ago2 around 44% to 56% [15, 29]. Our results agree with the previous studies, particularly in the group co-vaccinated with IFN-γ/D2, showing the dependence of IFN-γ effects for the positivity of anti-nRNP/Sm (Fig 1 Panel A).

When we measured by another assay such as ELISA, the same antibodies anti-nRNP/Sm and anti-Su/Ago2 in the experimental group D2/IL-10 were lower than those of the D2/IFN-γ co-vaccinated group suggesting an antigen-specific process induced by IL-10/D2 co-vaccination.

In our study, the renal histopathology showed minor kidney damage in the D2/IL-10 vaccinated group, showing glomerular cellularity and immune complex deposits than the D2/IFN-γ co-vaccinated mice. Also, a potential beneficial role of IL-10 in preventing immune complex glomerulonephritis (proteinuria).

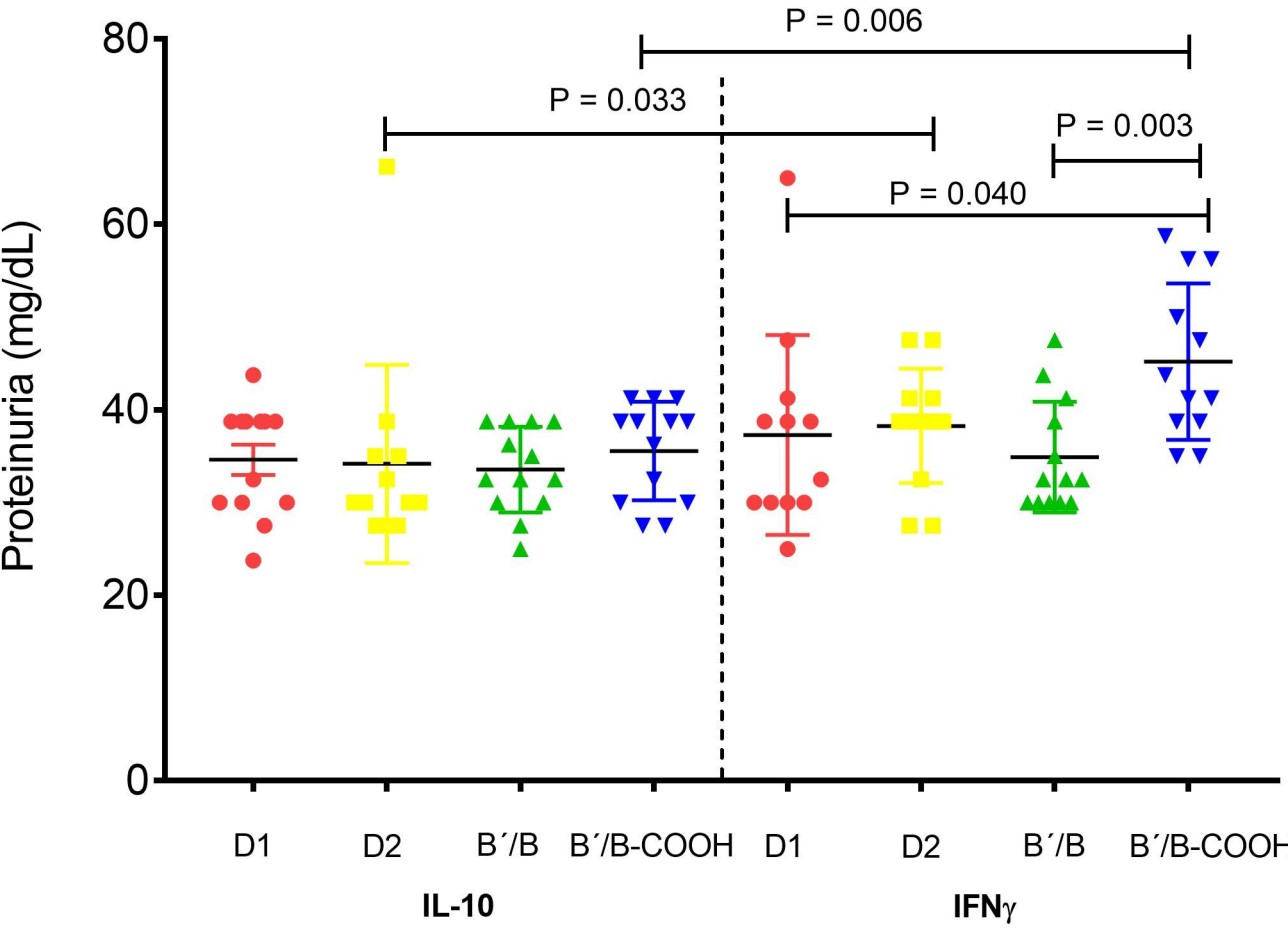

**Fig 5. Prevalence of proteinuria in pristane-treated mice vaccinated with Sm antigens co-vaccinated with IL-10 *vs.* IFN-γ.** Prevalence of proteinuria in pristane-treated mice vaccinated with Sm antigens co-vaccinated with IL-10 *vs.* IFN-γ. Each symbol represents one mouse. Mann-Whitney test was used.

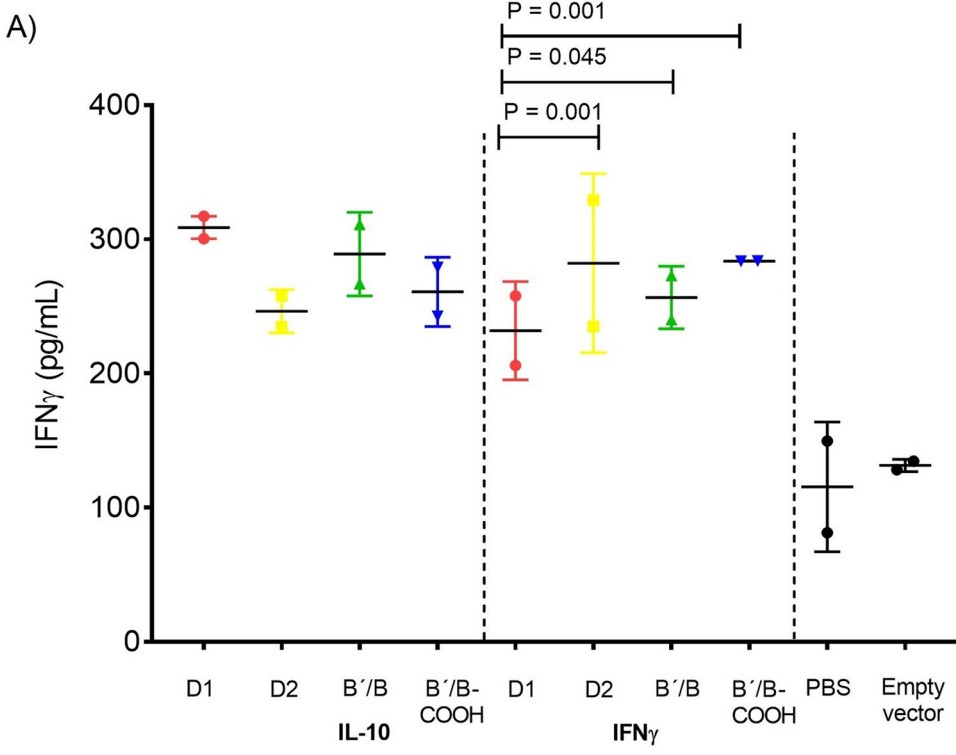

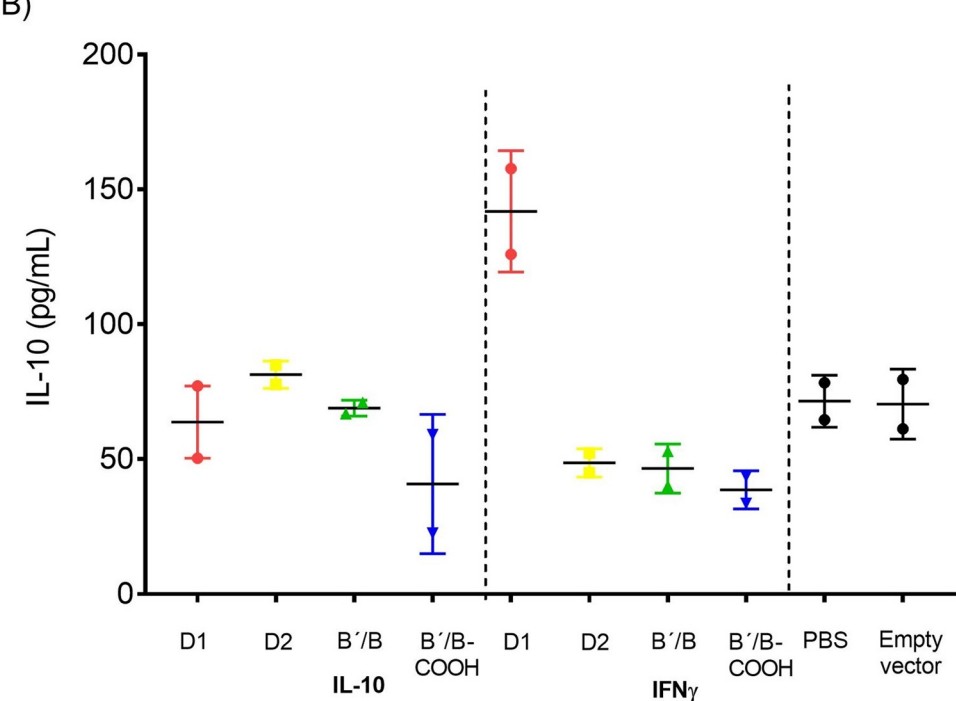

**Fig 6. Cytokine levels in splenocyte supernatants.** Cytokine levels in splenocyte supernatants. Panel A) IFN-γ levels in prophylactic co-vaccinated groups with $D1_{(83-119)}$, D2, B´B and $B´B_{COOH}$ along IFN-γ and IL-10, PBS and mock treatment. Panel B) IL-10 levels in prophylactic co-vaccinated groups with $D1_{(83-119)}$, D2, B´B and $B´B_{COOH}$ along IFN-γ and IL-10, PBS and mock treatment. Mann-Whitney test was used.

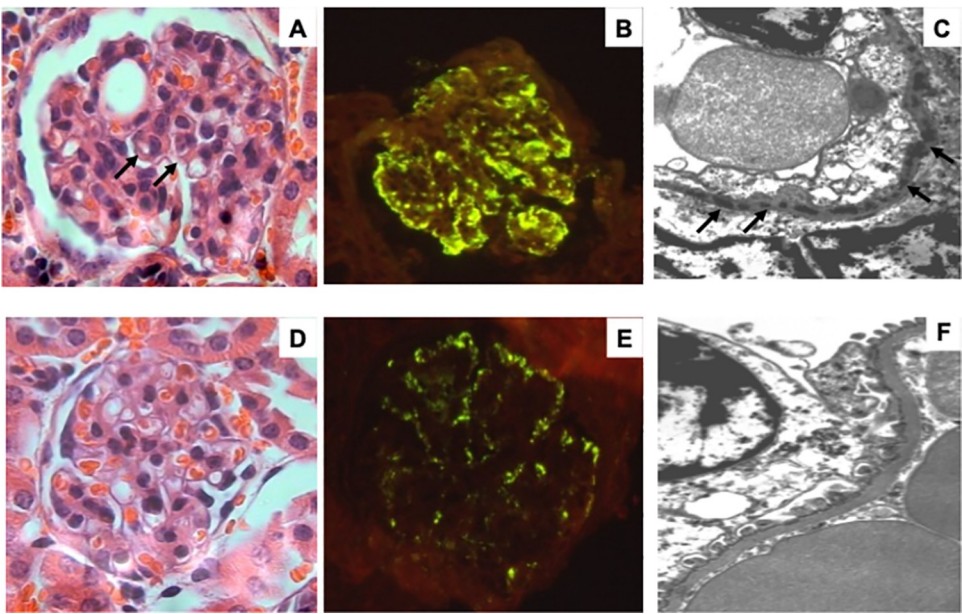

**Fig 7. Histology of kidneys from pristane-treated mice co-vaccinated with D2/IL-10 or D2/IFN-γ.** Representative histological and ultrastructural changes in the kidney from female *BALB/c* mice vaccinated with D2/IL-10 or D2/IFN-γ. Panel A) Light micrograph from IFN-γ/D2 co-vaccinated mouse showing a hypercellular glomerulus with mesangial matrix widening and thick capillary walls (arrows). Panel B) Immunofluorescence photograph taken from the same mouse (A) shows numerous mesangial and intramembranous IgG immune complexes. Panel C) Abundant electron-dense intramembranous deposits (arrows) are showed in this electron microscopy micrograph taken from the same IFN-γ/D2 prophylactic co-vaccinated mouse. Panel D) In contrast, lower cellularity and thinner capillary membranes are seen in this glomerulus from IL-10/D2 prophylactic co-vaccinated mouse. Panel E) The glomeruli from the same IL-10/D2 co-vaccinated mouse showed lesser immune complexes in the immunofluorescence. Panel F) in electron microscopy study (all light and fluorescence micrographs 100x, electron microscopy micrographs 25,000x).

It has been observed that the administration of autoantigens and functional cytokine genes with the employment of DNA expression vector has the potential to restore immune tolerance and develop a suppressive immunization could be established, and $T_H2$ shift with promising results [30, 37]. Experimental immunization against antigens that trigger autoimmune disorders caused by $T_H1$ auto-reactive cells such as multiple sclerosis (MS), type 1 diabetes, and RA has been developed with a promising result [21, 37–39]. In the case of EAE, a model of MS, studies demonstrate that the delivery of DNA vaccine with the self-peptide $PLP_{139-151}$ and IL-4 protected against induction of the disease induced by $PLP_{139-151}$ and caused that the antigen-specific autoreactive T cells shift their phenotype to a protective $T_H2$-type response [37]. These findings reinforced the results of a reduced spread of autoantibodies responses to epitopes on various myelin molecules with the use of a cocktail of tolerizing DNA vaccines encoding the four principal components of myelin, myelin basic protein (MBP), myelin-associated glyco-protein (MAG), myelin oligodendrocyte glycoprotein (MOG) and PLP with the addition of a plasmid encoding IL-4 [40].

In the pristane model of lupus, the inflammatory cytokines such as IFN-α, β and γ, IL-6, and IL-12 stimulate autoantibodies. In contrast, the deficiency of IFN-γ has been shown to have a protective effect on renal disease and autoantibodies production [31, 41, 42]. Studies in SLE patients suggest that interferon-related gene expression and pathways are standard features for SLE pathogenesis, where IFN-γ plays a critical role at SLE onset [17]. On the other hand, IL-10 is a regulatory cytokine that inhibits $T_H1$ cytokine production and proliferation of CD4+ T cells via its indirect effects on antigen-presenting cells (APC) function or through

direct effects on T cells. This cytokine is an essential modulator of disease activity in human SLE, where patients with lupus produce large amounts of IL-10 correlating with disease activity. Notwithstanding, studies in MRL-Fas$^{lpr}$ mice revealed that during the initial phases of the disease, IFN-γ and its induced IgG2a promote autoimmunity, and IL-10 appears to be needed to suppress such pathogenic $T_H1$ responses, including IFN-γ-mediated autoantibody production and renal inflammation [20].

The employment of DNA plasmids as a vector for immunization has some advantages over traditional vaccine modalities in terms of effectiveness, safety, and cost. One of the principal advantages is the minimal possibility integrating DNA vaccine to host chromosome and the anti-vector autoimmunity [43]. Nevertheless, there are some particularities of DNA vaccination mechanisms that we must consider and are still under investigation. For initiating the immune response by DNA vaccination, somatic cells (myocytes or keratinocytes) and professional antigen-presenting cells (APCs) are required. Subsequently, the plasmid-encoded gene entering the nucleus of transfected cells and foreign antigens are generated and processed by the major histocompatibility complex (MHC) class I or II of APC to prime naïve T cells in the draining lymph nodes [43]. To elicit a successful transfection, plasmid DNA is synthesized with the addition of viral promoters, transcriptional trans-activators, and enhancer elements that increase the transcription activity [43]. In our experiments, we constructed DNA vaccines with DNA plasmid that contain an active cytomegalovirus immediate early promoter (pCMV), consider a strong and ubiquitously viral promoter to control transgene expression [44, 45]. There have been observed that naked plasmid DNA controlled by the CMV promoter elicited a mixed $T_H1/T_H2$ response, followed by the compartmentalization of the immune response $T_H1$-like in the spleen and $T_H2$-biased in the draining lymphatic node [45]. Following intramuscular and intradermal immunization, the humoral response is characterized by the predominant production of specific IgG2a antibodies, whereas other inoculation methods such as gene guns yield a preponderance of IgG1 antibodies, suggesting $T_H1$ and $T_H2$-biased responses, respectively [45]. In our experiments, we were not able to determine the production of specific IgG1/IgG2a antibodies in mice treated with PBS and naked DNA plasmid (empty vector). However, we did not observe differences between IL-10 and IFN-γ levels among these groups.

The study presents some limitations; our design was a prophylactic experimental model that implied six-month period of treatment and sacrifice at the end of the trial. The number of biological serum samples was limited, and we used them until run out since multiple experiments of immunoprecipitation, autoantibodies measurement, etc. The kidney tissue obtained was analyzed with Mexico City; however, the unit hospital was reconverted to a COVID19 one, and our collaborator was retired. In this sense, in our experimental conditions was not feasible to do an NIH renal score activity and chronicity or to develop more immunofluorescence slides, nor cytokine determination as you requested.

To our knowledge, this is the first report about the employment of DNA vaccination with Sm antigens in combination with IL-10 in a pristane-induced lupus model. Our findings suggest that the co-vaccination of D2 plus IL-10 seems to produce less anti-nRNP/Sm autoantibodies when is administrated in a prophylactic manner. Further experiments are needed to reinforce our results, specifically the role of D2 in pristane-induced lupus.

## Conclusion

DNA co-vaccination cocktail of D2 plus IL-10 in pristane-induced lupus *BALB/c* mice use in a prophylactic way, improves the renal damage by acting as tolerizing therapy corroborated by the decrease in the levels of proteinuria and anti-nRNP/Sm.

## Supporting information

**S1 File. Immunoprecipitation gel images.**
(PDF)

**S2 File. The ARRIVE guidelines 2.0: Author checklist.**
(PDF)

**S1 Fig. Histology of kidneys from control and prophylactic DNA vaccination in a murine lupus model.**
(TIF)

## Acknowledgments

The authors would like to thank Carlos Alberto Franco Salazar for manuscript English edition.

## Author Contributions

**Conceptualization:** Beatriz Teresita Martín-Márquez, Erika Aurora Martínez-García, Flavio Sandoval-García, Monica Vázquez-Del Mercado.

**Data curation:** Minoru Satoh, Monica Vázquez-Del Mercado.

**Formal analysis:** Minoru Satoh, Fernanda Isadora Corona-Meraz.

**Funding acquisition:** Monica Vázquez-Del Mercado.

**Investigation:** Beatriz Teresita Martín-Márquez, Marcelo Heron Petri, Trinidad García-Iglesias, Monica Vázquez-Del Mercado.

**Methodology:** Beatriz Teresita Martín-Márquez, Minoru Satoh, Rogelio Hernández-Pando, Erika Aurora Martínez-García, Marcelo Heron Petri, Flavio Sandoval-García, Oscar Pizano-Martinez, Trinidad García-Iglesias, Monica Vázquez-Del Mercado.

**Resources:** Monica Vázquez-Del Mercado.

**Supervision:** Oscar Pizano-Martinez.

**Writing – original draft:** Beatriz Teresita Martín-Márquez, Monica Vázquez-Del Mercado.

**Writing – review & editing:** Erika Aurora Martínez-García, Flavio Sandoval-García, Fernanda Isadora Corona-Meraz.

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
