## [Decision Letter · Decision Letter 0]

13 Jul 2021

PONE-D-21-18801

The DNA co-vaccination using Sm antigen and IL-10 as prophylactic experimental  therapy, ameliorates nephritis in a model of lupus  induce by pristane

PLOS ONE

Dear Dr. Vázquez-Del Mercado,

Thank you for submitting your manuscript to PLOS ONE. After careful consideration, we feel that it has merit but does not fully meet PLOS ONE’s publication criteria as it currently stands. Therefore, we invite you to submit a revised version of the manuscript that addresses the points raised during the review process.

Our reviewers found some merits and interests in this study, but also pointed out a number of criticisms that require improvement or amendment. Additional experiments are necessary to answer points raised by reviewers. I ask authors to fully respond to all comments in the revised version. 

We look forward to receiving your revised manuscript.

Kind regards,

Masataka Kuwana, MD, PhD

Academic Editor

PLOS ONE

Journal Requirements:

4. To comply with PLOS ONE submissions requirements, in your Methods section, please provide additional information on the animal research and ensure you have included details on efforts to alleviate suffering.

5. We suggest you thoroughly copyedit your manuscript for language usage, spelling, and grammar. If you do not know anyone who can help you do this, you may wish to consider employing a professional scientific editing service.  

6. As part of your revision, please complete and submit a copy of the Full ARRIVE 2.0 Guidelines checklist, a document that aims to improve experimental reporting and reproducibility of animal studies for purposes of post-publication data analysis and reproducibility: https://arriveguidelines.org/sites/arrive/files/Author%20Checklist%20-%20Full.pdf (PDF). Please include your completed checklist as a Supporting Information file. Note that if your paper is accepted for publication, this checklist will be published as part of your article.

Reviewers' comments:

Reviewer's Responses to Questions

**Comments to the Author**

1. Is the manuscript technically sound, and do the data support the conclusions?

Reviewer #1: No

Reviewer #2: Partly

2. Has the statistical analysis been performed appropriately and rigorously? 

Reviewer #1: Yes

Reviewer #2: No

3. Have the authors made all data underlying the findings in their manuscript fully available?

Reviewer #1: Yes

Reviewer #2: No

4. Is the manuscript presented in an intelligible fashion and written in standard English?

Reviewer #1: Yes

Reviewer #2: No

5. Review Comments to the Author

Reviewer #1: This is an interesting original and novel study that observing the role of Sm-antigen vaccination with IL-10 to induce tolerogenic immunity against lupus. However, several things need to be reworked to obtain a clear conclusion:

1. The vaccination of D2/IL-10 was able to reduce the anti-nRNP/Sm titers, but it was still not known whether this reduction was caused by the D2 or IL-10. The IL-10 alone was already known that could decrease antibody productions due to its immunoregulatory properties. Therefore, the authors should also observe the comparison between the antibody titers after the administration of D2 or IL-10 alone, and the combinations of D2/IL-10.

2. Proteinuria was measured in this study by using a dipstick examination. Rather than using a dipstick examination, which was a semi-qualitative examination, I suggest using quantitative albumin creatinine ratio (ACR) or protein creatinine ratio (PCR) examinations that had better power to explain the extent of proteinuria.

3. The renal pathology was explained in a qualitative approach. Although the authors explained several improvements of the renal pathology findings after the vaccinations, there should be some quantitative measurement that could explain the result more objectively. Several scoring could be used to assess the severity of lupus nephritis, such as NIH Lupus Nephritis Activity and Chronicity Indices.

Reviewer #2: In this manuscript the authors present data suggesting that prophylactic co-vaccination with sm antigen peptides and IL-10 exert a protective role in pristane-induce lupus in Balb/c mice.

There are several issues with the paper that needs to be addressed.

1. It is imperative that all data are shown. For example, figure 2 needs to show data from individual mice as does the bargraphs in figures 4 and 5. Please follow the same standard as used in figure 3.

2. To help the reader, please provide a schematic overview of the animal studies including at what time the mice were injected with what and how many per group. While the information can be pieced together from the materials and methods section it is not easy to read and understand. this is particularly true as only subsets of mice were used for end-of-study analyses (3 mice for kidney analyses, 4 mice for spleen analyses, etc.). Please identify why all studies were not done for all mice, why and how mice for these studies were selected etc.? If the mice came from individual setups (i.e. all mice were not treated with pristane and vaccine at the same time), please indicate each individual cohort of mice and make it clear to the reader when all mice were analyzed together.

3. Please expand the parts of the materials and methods sections that are listed with only a reference. A brief description of the protocol and the specific dilutions used should be written in the manuscript.

4. For proteinuria data, please show individual data as the current bar graphs misleadingly shows that the measure is exact, not based on a dipstick analysis.

5. For kidney analysis, it is unclear if half kidneys were preserved in any solvent/gel such as OCT or the like. As written, the kidneys appear to have been frozen directly. Such protocol would induce water crystallization and tissue damage.

6. Figure 1 shows the presence of specific autoantibodies in sera of co-vaccinated mice. The gel plots do not show recombinant proteins to verify the different specificities. Please include. The gel plots do not show any size ladder to verify protein sizes. Please include. There are 10 samples shown in A and 9 samples shown in B, however, in materials and methods section, it is written that 13 mice were used per group. Please explain or include the remaining data.

7. It is assumed that the percentages given in lines 242-244 and 254-257 are a result of the gel images in figure 1. How is it determined that there is no difference between 54% and 10% as stated? It would be helpful if the authors could present not just the gel plots, but also a graph representing the obtained data.

8. Several sets of data are not supported by the data shown. for example, the authors claim that there are higher levels of IgG2a/IgG1 in mice co-vaccinated with IFNg than in mice co-vaccinated with IL-10 (line 265-267), however, there are no statistical differences between the groups (figure 3). Make sure not to overinterpret the data here and in the discussion.

9. Based on the bargraph appearance of proteinuria data in figure 4 and the highly overlapping error bars, this reviewer is surprised to see statistical significance. Please explain clearly in the figure legend what statistical method was used. As mentioned above, please show individual values rather than the mean/stdev.

10. Figure 5 needs to be redone so that the control samples are represented along with the pristane-treated samples. Please move figure 5A, IFNg-measurements to figure 5B and figure 5A, IL-10 measurements to figure 5C. This will facilitate better understanding of the enhanced production of each cytokine in response to the co-vaccination.

11. The renal histology is very nice and well represented. For comparison, it would be helpful if the authors could solicit a renal score by a renal pathologist blinded to the treatment groups. Also, please provide images from all the mice done - if not in the primary figure, then in a supplemental figure.

12. Similarly, it would be very helpful if the authors could quantify the immunofluorescence stainings either by intensity or by area, underscoring the reduced deposition of immune complexes in IL-10 covaccinated animals.

13. In the discussion the authors write: "We observed that the IgG1, IgG2a levels and IgG2a/IgG1 ratio showed that the prophylactic co-vaccination influences the TH1/TH2 polarization during the course of the disease." This appears to be an overinterpretation of the data as no statistical difference between IgG2a/IgG1 ratios were found. Furthermore, the authors do not provide any evidence for changes in the Th1/Th2 polarization. To support this statement, please provide evidence for altered Th1/Th2 cell levels using either master transcription factors or intracellular cytokines to determine each cell subset.

14. The manuscript needs proof reading for missing words, wrong syntax an typographical errors. If needed, please consult a native English speaker.

-

6. PLOS authors have the option to publish the peer review history of their article (what does this mean?). If published, this will include your full peer review and any attached files.

Reviewer #1: No

Reviewer #2: No

---

## [Author Response · Author response to Decision Letter 0]

8 Sep 2021

PONE-D-21-18801

The DNA co-vaccination using Sm antigen and IL-10 as prophylactic experimental therapy ameliorates nephritis in a model of lupus induced by pristane

Response to Reviewers

Reviewer #1

This is an interesting original and novel study that observing the role of Sm-antigen vaccination with IL-10 to induce tolerogenic immunity against lupus. However, several things need to be reworked to obtain a clear conclusion:

1. The vaccination of D2/IL-10 was able to reduce the anti-nRNP/Sm titers, but it was still not known whether this reduction was caused by the D2 or IL-10. The IL-10 alone was already known that could decrease antibody productions due to its immunoregulatory properties. Therefore, the authors should also observe the comparison between the antibody titers after the administration of D2 or IL-10 alone, and the combinations of D2/IL-10.

R= We appreciate your valuable comments and suggestions for further experiments. First of all, the design of our study was based on previous literature and reliable data dealing about tolerizing therapy. It is already known that the region of Sm antigen spanning the aa 83-199 also known as D1, includes a second region identified as Sm D2 antigen spanning the aa 90-102. Why is important these concepts? It has been demonstrated the tolerogenic effect of Sm D1 combined or in cocktail with IL-10 DNA vaccination delays autoantibody production and prolonged survival in lupus mice (PMID 15494537). 

2. Proteinuria was measured in this study by using a dipstick examination. Rather than using a dipstick examination, which was a semi-qualitative examination, I suggest using quantitative albumin creatinine ratio (ACR) or protein creatinine ratio (PCR) examinations that had better power to explain the extent of proteinuria.

R= In fact you are absolutely right, however this approach would imply a whole new experiment that is not able to do for the time that requires (more than 6 months). Notwithstanding, we decided to answer in the best way possible the suggestions of the academic and reviewers. In this sense, we showed in Figure 5, the semiquantitative proteinuria in such a way that you could see statistical differences by group of mice. 

Fig 5. Prevalence of proteinuria in pristane-treated mice vaccinated with Sm antigens co-vaccinated with IFN-� vs. IL-10. Proteinuria was less common in IL-10 co-vaccinated group vs. IFN-� group for the same Sm antigen. 

3. The renal pathology was explained in a qualitative approach. Although the authors explained several improvements of the renal pathology findings after the vaccinations, there should be some quantitative measurement that could explain the result more objectively. Several scoring could be used to assess the severity of lupus nephritis, such as NIH Lupus Nephritis Activity and Chronicity Indices.

R= We appreciate your comments. The type of this study is experimental and the analysis of variables was cross-sectional at the end of a 6-month period. In this sense, we did not measure a treatment response or disease progression, that is why the NIH Lupus Nephritis Activity and Chronicity Indices were not assessed. Besides, we asked for the embedded paraffin tissues from our experimental groups to our collaborator in Mexico City, however for the pandemia SARS-Cov2 and the retirement of the pathologist was not possible to rescue them in order to do the NIH score you requested.

Reviewer #2

In this manuscript the authors present data suggesting that prophylactic co-vaccination with Sm antigen peptides and IL-10 exert a protective role in pristane-induce lupus in Balb/c mice.There are several issues with the paper that needs to be addressed.

1. It is imperative that all data are shown. For example, figure 2 needs to show data from individual mice as does the bargraphs in figures 4 and 5. Please follow the same standard as used in figure 3.

R= Done.

Fig 3. Effects of prophylactic co-vaccination of Sm antigens with IL-10 or IFN-g on anti-nRNP/Sm and anti-Su/Ago2 levels. Sera were tested for A) anti-nRNP/Sm and B) anti-Su/Ago2 antibodies by ELISA at 6 months after prophylactic co-vaccination and induction of lupus by pristane. 

Fig 5. Prevalence of proteinuria in pristane-treated mice vaccinated with Sm antigens co-vaccinated with IFN-� vs. IL-10. Proteinuria was less common in IL-10 co-vaccinated group vs. IFN-� group for the same Sm antigen. 

Fig 6. Cytokine levels in splenocyte supernatants. A) IFN-� and IL-10 levels in PBS and mock treatment mice. B) IFN-� levels in prophylactic co-vaccinated groups with D183-119, D2, B´B and B´BCOOH along IFN-� and IL-10 C) IL-10 levels in prophylactic co-vaccinated groups with D183-119, D2, B´B and B´BCOOH along IFN-� and IL-10.

2. To help the reader, please provide a schematic overview of the animal studies including at what time the mice were injected with what and how many per group. While the information can be pieced together from the materials and methods section it is not easy to read and understand. this is particularly true as only subsets of mice were used for end-of-study analyses (3 mice for kidney analyses, 4 mice for spleen analyses, etc.). Please identify why all studies were not done for all mice, why and how mice for these studies were selected etc.? If the mice came from individual setups (i.e. all mice were not treated with pristane and vaccine at the same time), please indicate each individual cohort of mice and make it clear to the reader when all mice were analyzed together.

R= We added Figure 1, where we described plenty in detail all the experimental procedure as you will find in Material and Methods section.

3. Please expand the parts of the materials and methods sections that are listed with only a reference. A brief description of the protocol and the specific dilutions used should be written in the manuscript.

R= We appreciate your comments. In the Material and Methods section, we added the lines 194-204 as follows:

Immunoprecipitation:

The presence of autoantibodies was tested by immunoprecipitation of 35S-methionine labeled K562 (human erythroleukemia) cell extract and SDS-PAGE as described. Briefly, the cells were labeled with [35S] methionine (DuPont-New England Nuclear, Boston, MA), lysed in NET/NP-40 buffer (150 mM NaCl, 2mM EDTA, 50 mM Tris-HCl pH 7.5, 0.3% NP-40), containing 0.5 mM PMSF, 0.3 TIU/mL aprotinin, and immunoprecipitated using protein A-Sepharose beads (Pharmacia LKB Biotechnology, Inc, Piscataway, NJ) coated with 5 �L of mouse sera plus 12 �L of rabbit anti-mouse IgG1 (1 mg/mL). After, several washes were done using 0.5 M NaCl NET/NP40 (0.5 M NaCl, 2 mM EDTA, 50 mM Tris-HCl, pH 7.5, 0.3% NP-40). American Type Culture Collection, Rockville, MD). Immunoprecipitated proteins were analyzed by autoradiography.

Lines 222-233:

Anti-Su/Ago2 ELISA:

The determination of anti-Su/Ago2 by ELISA was as described. Wells of microtiter plates (Nunc, Immobilizer Amino™) were coated with 50 �L of human anti-Su IgG at 20 �g / mL in 20 mM Tris-HCl pH 8.0. Half of the wells were incubated for 1 h at 22º C with 50 �L of K562 cell lysate and the other half with NET/NP-40 alone, 100 �L of diluted mouse serum (1/250 in blocking solution) was added to wells coated with either K562 cell lysate of buffer alone, and incubated for 1 h at room temperature. The wells were then washed using with NET/NP-40, and incubated with alkaline phosphatase-conjugated goat anti-mouse IgG antibodies (Southern Biotechnology, Birmingham, AL), later developed with 1 mg / 1mL p-nitrophenyl phosphate substrate (Sigma) in diethanolamine buffer (1 M diethanolamine-HCl pH 9.6, 0.5 nM MgCl2, 0.02% NaN3) and OD was read at 405 nm in an automated ELISA reader.

Lines 235-243:

Measurement of immunoglobulin (IgG) levels:

Total levels of each Ig isotype were determined by ELISA coated with 50 �L/well of 3 �g/mL of goat anti-mouse �/� light chain antibodies (9:1 ratio, Southern Biotechnology, Birmingham, AL). Wells were washed with 0.15 M NaCl, 2 mM EDTA, 50 mM Tris containing 0.3% (NET/NP-40) and then blocked with 0.5% BSA in NET/NP-40. Murine sera were diluted 1:200,000 with NET/NP-40 containing 0.5% BSA. The plate was washed and incubated with 1:1000 dilution of alkaline phosphatase-labeled goat anti-mouse antibodies specific for IgG1 and IgG2a (Southern Biotechnology). Read at 405 nm using an ELISA plate reader.

4. For proteinuria data, please show individual data as the current bar graphs misleadingly shows that the measure is exact, not based on a dipstick analysis.

R= Please refer to answer 2 of the response to Reviewer 1.

5. For kidney analysis, it is unclear if half kidneys were preserved in any solvent/gel such as OCT or the like. As written, the kidneys appear to have been frozen directly. Such protocol would induce water crystallization and tissue damage.

R= We appreciate your valuable comments, however we did not cause a crystallization effect. The detail methodology employed is described on lines 271-285:

Renal pathology:

Immediately after euthanized the mice, both kidneys from three animals per experimental group were removed and divided into two halves. For immunofluorescence study, one-half was frozen by immersion in liquid nitrogen and sectioned at 5 microns thickness, incubated with FITC-conjugated rabbit anti-mouse IgG antibodies, and analyzed under an epifluorescence microscope. From the other kidney half, we obtained a thin layer of 1mm from the cortex, fragmented into small tissue pieces immediately fixed by immersion in 4% glutaraldehyde dissolved in cacodylate buffer (Sigma Aldrich�) for 2hr at 4°C. After washing, kidney tissue fragments were dehydrated in graded ethyl alcohols and embedded in Epon resin. Sections of 1-micron thickness were stained with toluidine blue and used to select representative tissue areas from which thin sections were obtained. Thin tissue sections were mounted in copper grids, stained with citrate lead and uranium salts and examined in a Zeiss EM 10 electron microscopy. The rest of the kidney tissue was fixed by immersion in 10% formaldehyde dissolved in PBS, dehydrated, and embedded in paraffin for conventional histology. 

6. Figure 1 shows the presence of specific autoantibodies in sera of co-vaccinated mice. The gel plots do not show recombinant proteins to verify the different specificities. Please include. The gel plots do not show any size ladder to verify protein sizes. Please include. There are 10 samples shown in A and 9 samples shown in B, however, in materials and methods section, it is written that 13 mice were used per group. Please explain or include the remaining data.

R= Unfortunately, by the time we did the immunoprecipitation, some samples were run out since the measurements of autoantibodies etc. 

7. It is assumed that the percentages given in lines 242-244 and 254-257 are a result of the gel images in figure 1. How is it determined that there is no difference between 54% and 10% as stated? It would be helpful if the authors could present not just the gel plots, but also a graph representing the obtained data.

R= Done

8. Several sets of data are not supported by the data shown. for example, the authors claim that there are higher levels of IgG2a/IgG1 in mice co-vaccinated with IFN-g than in mice co-vaccinated with IL-10 (line 265-267), however, there are no statistical differences between the groups (figure 3). Make sure not to overinterpret the data here and in the discussion.

R=Indeed, as you commented, since there is no significant difference, we cannot affirm that there are higher levels of IgG2a in the groups vaccinated with IFN-γ than in those with IL-10. So, we have corrected the wording in these paragraphs in results:

IgG1, IgG2a subclass concentration and IgG2a/IgG1 ratio: 

We did not obtain statistically significant differences in the levels of IgG1, IgG2a and IgG2a / IgG1 ratio between the groups vaccinated with IFN-g and IL-10.

9. Based on the bargraph appearance of proteinuria data in figure 4 and the highly overlapping error bars, this reviewer is surprised to see statistical significance. Please explain clearly in the figure legend what statistical method was used. As mentioned above, please show individual values rather than the mean/stdev.

R= We appreciate your observation. The graphic was emended and the legend describe the use of Mann-Whitney test.

10. Figure 5 needs to be redone so that the control samples are represented along with the pristane-treated samples. Please move figure 5A, IFNg-measurements to figure 5B and figure 5A, IL-10 measurements to figure 5C. This will facilitate better understanding of the enhanced production of each cytokine in response to the co-vaccination.

R= Done

11. The renal histology is very nice and well represented. For comparison, it would be helpful if the authors could solicit a renal score by a renal pathologist blinded to the treatment groups. Also, please provide images from all the mice done - if not in the primary figure, then in a supplemental figure.

R= We asked for the embedded paraffin tissues from our experimental groups to México city, Instituto de Ciencias Médicas y Nutrición Salvador Zubirán, however for the pandemia and the retirement of the pathologist it was not possible to rescue them in order to do a score as you requested. All figures we have were sent as supplemental material available at: figshare.com/articles/online_resource/PONE-D-21-18801/16441683.

12. Similarly, it would be very helpful if the authors could quantify the immunofluorescence stainings either by intensity or by area, underscoring the reduced deposition of immune complexes in IL-10 covaccinated animals.

R= Since you know, the direct immunofluorescence done is no longer available because the time spent and unfortunately tissues from our experimental groups that were stored in the Instituto de Ciencias Médicas y Nutrición Salvador Zubirán, are no longer available for the retirement of the pathologist since pandemic administrative health protocols. 

13. In the discussion the authors write: "We observed that the IgG1, IgG2a levels and IgG2a/IgG1 ratio showed that the prophylactic co-vaccination influences the TH1/TH2 polarization during the course of the disease." This appears to be an overinterpretation of the data as no statistical difference between IgG2a/IgG1 ratios were found. Furthermore, the authors do not provide any evidence for changes in the Th1/Th2 polarization. To support this statement, please provide evidence for altered Th1/Th2 cell levels using either master transcription factors or intracellular cytokines to determine each cell subset.

R= Indeed, as you comment, since there is no significant difference, we cannot affirm that there are higher levels of IgG2a in the groups vaccinated with IFN-g than in those with IL-10. So, we have deleted this sentence in the discussion section.

On the other hand, we are not able to do more experiments since the supernatant from splenocytes culture and serum stored are not useful to do these types of experiments. We will need to design other whole experiment to obtain new material for experimentation, especially serum samples and or splenocyte cultures.

14. The manuscript needs proof reading for missing words, wrong syntax an typographical errors. If needed, please consult a native English speaker.

R= The English edition of the manuscript was done by Carlos Alberto Franco Salazar.

---

## [Decision Letter · Decision Letter 1]

13 Oct 2021

The DNA co-vaccination using Sm antigen and IL-10 as prophylactic experimental therapy ameliorates nephritis in a model of lupus induced by pristane.

PONE-D-21-18801R1

Dear Dr. Vázquez-Del Mercado,

We’re pleased to inform you that your manuscript has been judged scientifically suitable for publication and will be formally accepted for publication once it meets all outstanding technical requirements.

Kind regards,

Masataka Kuwana, MD, PhD

Academic Editor

PLOS ONE

Additional Editor Comments (optional):

Reviewers' comments:

Reviewer's Responses to Questions

**Comments to the Author**

1. If the authors have adequately addressed your comments raised in a previous round of review and you feel that this manuscript is now acceptable for publication, you may indicate that here to bypass the “Comments to the Author” section, enter your conflict of interest statement in the “Confidential to Editor” section, and submit your "Accept" recommendation.

Reviewer #2: (No Response)

2. Is the manuscript technically sound, and do the data support the conclusions?

Reviewer #2: Yes

3. Has the statistical analysis been performed appropriately and rigorously? 

Reviewer #2: Yes

4. Have the authors made all data underlying the findings in their manuscript fully available?

Reviewer #2: Yes

5. Is the manuscript presented in an intelligible fashion and written in standard English?

Reviewer #2: Yes

6. Review Comments to the Author

Reviewer #2: Thank you for responding to my previous comments. The insertion of figure 1 is very helpful. Only a single comment left.

Please make sure that all figure legends adequately describe their figures. this include a basic description of the mice, a notion that each symbol represents one mouse, the n for each group and the statistical method used. This is done for most legends now, but is still missing from a few.

7. PLOS authors have the option to publish the peer review history of their article (what does this mean?). If published, this will include your full peer review and any attached files.

Reviewer #2: No

---

## [Editor Report · Acceptance letter]

18 Oct 2021

PONE-D-21-18801R1 

The DNA co-vaccination using Sm antigen and IL-10 as prophylactic experimental therapy ameliorates nephritis in a model of lupus induced by pristane. 

Dear Dr. Vázquez-Del Mercado:

I'm pleased to inform you that your manuscript has been deemed suitable for publication in PLOS ONE. Congratulations! Your manuscript is now with our production department. 

Kind regards, 

on behalf of

Prof. Masataka Kuwana 

Academic Editor

PLOS ONE